# Antimicrobial Peptides (AMPs) in the Pathogenesis of Alzheimer’s Disease: Implications for Diagnosis and Treatment

**DOI:** 10.3390/antibiotics11060726

**Published:** 2022-05-28

**Authors:** Francesco Bruno, Antonio Malvaso, Sonia Canterini, Amalia Cecilia Bruni

**Affiliations:** 1Regional Neurogenetic Centre (CRN), Department of Primary Care, ASP Catanzaro, 88046 Lamezia Terme, Italy; 2Association for Neurogenetic Research (ARN), 88046 Lamezia Terme, Italy; amaliaceciliabruni@gmail.com; 3Neurology Unit, IRCCS San Raffaele Scientific Institute, 20132 Milan, Italy; malvaso.antonio@libero.it; 4Division of Neuroscience, Department of Psychology, University La Sapienza, 00158 Rome, Italy; sonia.canterini@uniroma1.it

**Keywords:** antimicrobial peptides (AMPs), Alzheimer’s disease (AD), infectious hypothesis, beta-amyloid (Aβ), lactoferrin, defensins, cystatins, thymosin β4, histatin 1, statherin

## Abstract

Alzheimer’s disease (AD) represents the most frequent type of dementia in elderly people. There are two major forms of the disease: sporadic (SAD)—whose causes are not completely understood—and familial (FAD)—with clear autosomal dominant inheritance. The two main hallmarks of AD are extracellular deposits of amyloid-beta (Aβ) peptide and intracellular deposits of the hyperphosphorylated form of the tau protein (P-tau). An ever-growing body of research supports the infectious hypothesis of sporadic forms of AD. Indeed, it has been documented that some pathogens, such as herpesviruses and certain bacterial species, are commonly present in AD patients, prompting recent clinical research to focus on the characterization of antimicrobial peptides (AMPs) in this pathology. The literature also demonstrates that Aβ can be considered itself as an AMP; thus, representing a type of innate immune defense peptide that protects the host against a variety of pathogens. Beyond Aβ, other proteins with antimicrobial activity, such as lactoferrin, defensins, cystatins, thymosin β4, LL37, histatin 1, and statherin have been shown to be involved in AD. Here, we summarized and discussed these findings and explored the diagnostic and therapeutic potential of AMPs in AD.

## 1. Introduction

Alzheimer’s disease (AD) is the most frequent type of dementia in elderly people [1]. The clinical features of AD include both cognitive decline and a set of non-cognitive symptoms involving perception, mood, personality, and basic functioning, overall known as Neuropsychiatric or Behavioral and Psychological Symptoms of Dementia (BPSD) [2,3]. Two major forms of the disease exist: sporadic (SAD)—whose causes are not completely understood—and familial (FAD)—with a clear autosomal dominant inheritance [4,5].

The neuropathology of AD is characterized by diffuse brain atrophy and a reduction in brain volume and weight by approximately 20%, compared to control people [6,7]. At the microscopic level, the main neuropathological features of AD are: (i) amyloid plaques which consist of extracellular deposits of amyloid beta (Aβ) peptide and other molecules associated with axonal and dendritic damage [8]; (ii) tangles or neurofibrillary aggregates, that are intracellular deposits of paired helical filaments (PHF). The major component of these filaments is the hyperphosphorylated form of the protein tau (P-tau). When tau is hyperphosphorylated, its ability to bind microtubules decreases and aggregates abnormally resulting in the formation of PHFs. The aggregation of P-tau into filaments leads to the collapse of microtubules and the reduction in axonal transport [9]. Other neuropathological features of AD are represented by synaptic and neuronal loss [10], neuroinflammation accompanied by reactive gliosis [11], and a neuronal accumulation of iron [12] and cytoplasmatic granulovacuolar degeneration bodies [13].

Several hypotheses have been formulated to explain how these neuropathological features are causally related to each other underpinning the pathogenesis of AD. The “amyloid cascade hypothesis” [14] postulates that the progressive accumulation of Aβ in the brain triggers a complex cascade of events that result in the loss of synapses, a progressive deficiency of neurotransmitters, and the death of neuronal cells. According to the “cholinergic hypothesis”, AD is the result of a primary degenerative process that selectively affects the cholinergic neurons of the brain regions that exert an important function of awareness, attention, learning, and memory such as the hippocampus, amygdala, basal nuclei, and medial septum [15]. The “inflammation hypothesis” considers that the inflammatory reaction is a downstream effect of the accumulation of Aβ and P-tau proteins [16]. Finally, the “*infectious or microbial hypothesis*” proposes that pathogens, such as viruses, bacteria, and prions, represent the main cause of AD [17]. In support of this hypothesis, it has been documented that some pathogens, such as herpesviruses and some bacterial species, are commonly present in AD patients [18] prompting recent clinical research to focus on the characterization of antimicrobial peptides (AMPs), as a novel frontier for the study of this pathology.

AMPs, most of which are also known as host defense peptides (HDPs) [19,20], represent a very heterogeneous class of low molecular weight peptides, consisting in most cases of 50–100 amino acids [20,21]. As suggested by Moir et al. [22], AMPs are abundant in the brain and other immune-privileged tissues. Indeed, AMPs play a key role in the so-called “innate or natural immunity”, consisting of a series of non-specific defense mechanisms directed towards a wide spectrum of microorganisms and present from birth [23]. These immune responses are pre-existing to exposure to the foreign substance (antigen) and represent the body’s first defense barrier to pathogens. However, their production can be also induced by inflammation [24,25].

Unlike classic antibiotics, AMPs are both ribosomally and non-ribosomally derived [26,27]. Based on their secondary structure, AMPs are commonly classified into α-helical, β-sheet, or peptides with extended/random-coil structure, with most of the AMPs belonging to the first two classes [20]. Likewise, based on their mechanism of action, AMPs are classified into membrane acting and non-membrane acting peptides. The first class of AMPs mainly harbor cationic peptides causing the disruption of the physical integrity of the microbial membrane [21]. Non-membrane acting peptides translocate into the cytoplasm of bacteria to act on intracellular targets [28,29].

These two modes of action do not allow bacteria to develop resistance, unlike what happens to conventional antibiotics [20,21]. AMPs also inhibit bacterial protein, nucleic acid, cell wall synthesis, and enzymatic activities [30,31,32]. In addition to bactericidal effects, AMPs are also antiviral, antifungal [33], antitumor [34], and immunomodulatory [22,35,36,37], and are involved in autoimmune diseases [38]. 

It has recently been proposed that Aβ can be considered an AMP; thus, representing a type of innate immune defense protein that protects the host from a variety of pathogens [22,39,40] and that other proteins with antimicrobial activity, such as α- and β-defensins, lactoferrin, cystatins A and B, histatin 1, statherin, and thymosin β4 play a key role in AD [41,42,43,44,45]. 

However, to our knowledge, the potential role of these AMPs in the pathogenesis of AD, as well as a tool to open new horizons in diagnosis and treatment of AD, have not been systematically reviewed and discussed. 

The purpose of this review is to comprehensively and critically analyze the current experimental evidence on this topic, suggesting significant issues for future studies that are then put forward.

## 2. The Infectious Hypothesis of AD 

Infection is a process characterized by the penetration and multiplication in living tissues of pathogenic microorganisms or viruses. The idea that infections could underlie AD was first proposed in 1907 by Oskar Fischer, Alois Alzheimer’s “rival” [46]. However, this hypothesis remained largely unexplored until 1991, when Jamieson et al. [47] found the Herpes Simplex Virus 1 (HSV-1) DNA in the brain of AD patients. Since then, many scientists have investigated the possible causal relationship between various pathogens (e.g., viruses, parasites, bacteria, fungi) and the onset of AD [48]. 

According to the infectious or pathogens hypothesis of sporadic AD, normal ageing is associated with a weakening of the brain–blood barrier (BBB) and immune system and infection with pathogenic viruses, bacteria, fungi, or parasites leads to chronic neuroinflammation which in turn promote the production and aggregation of Aβ and P-tau and consequently neuronal degeneration [48]. 

Over the past 30 years, various evidence has been collected to support this hypothesis: (i) the presence of several pathogens (e.g., viruses, parasites, bacteria, fungi) in the brain of most AD patients; (ii) the colocalization of pathogens with Aβ plaques in the brain of AD patients; and (iii) the transmissibility of key features through intracerebral injection of AD brain homogenates (for a review see: Vigasova et al. [48]; Sochocka et al. [49]). More recently, multi-microbial or poly-microbial hypothesis has also been proposed which postulates that the collective and cumulative activity of different pathogens (e.g., viruses and bacteria, bacteria and fungi, viruses, bacteria and fungi) contributes to the development of AD (for a review see Vigasova et al. [48]). 

Although no definitive conclusions can be made regarding a causal role for pathogens in AD, a dramatic reduction in dementia risk with anti-herpetic treatment has been shown [50,51] suggesting that pathogens could represent a powerful risk factor for the development of AD, and thus opening new and unexplored ways for the AD characterization, diagnosis, treatment, and prevention. 

## 3. AMPs Involvement in AD

### 3.1. Aβ

Aβ is a phylogenetically ancient peptide and highly conserved among species, although its physiological functions are not yet fully understood [52]. Several lines of evidence suggest that it can be considered a multifunctional peptide able to: (i) regulate learning, memory, and neurogenesis; (ii) promote blood–brain barrier repair following injury; and (iii) act as a tumor suppressor and AMP [53]. 

Aβ is derived from proteolytic processing of a transmembrane protein known as amyloid protein precursor (APP) [54]. APP can be metabolized in the cell according to two different processes. The first is the α-secretase and ADAM10 non-amyloidogenic pathway that leads, among others, to the formation of soluble sAPPα with neuroprotective properties [55]. The second is the β- and γ-secretase amyloidogenic pathway in which APP is firstly cleaved by β-secretase and then by γ-secretase. This pathway leads to the production of releasing the AD-associated protein Aβ_1-42_ which represents the main constituent of Aβ senile plaques [39,56]. 

Currently, more than 40 different Aβ peptide variants composed of 37–43 amino acids have been identified [57,58,59]. Although it represents the main component of senile plaques in AD, accumulation of Aβ has also been observed in the brains of healthy elderly subjects [60,61]. Furthermore, Aβ senile plaques were also detected in mice after intranasal infection with bacteria [62]. These data, together with the well-documented structural homologies with AMPs [40,63,64], suggested that Aβ may represent per se an AMP involved in the innate immune system, [39,63]. Within the framework of the infectious etiology of AD, the Amyloid Protection Hypothesis has recently been proposed which postulated that the Aβ peptide accumulation represents an innate immune response targeted to fight and neutralize infections rather than being the main factor responsible for the AD’s pathophysiology [22,65]. 

The idea that Aβ could be considered as a component of the innate immune system was first proposed in 2002 by Robinson and Bishop [66] in the “*Bioflocculant hypothesis*”. According to the authors, the aggregative properties of Aβ are due its ability to surround and sequester pathogens—in the brain to limit their spread and—at the same time—prepare phagocytosis. This hypothesis was supported by the identification of microbial DNA within Aβ senile plaques and the attraction of the positive charge of Aβ by the negatively charged membrane of pathogens [66,67]. A few years later, a low production of Aβ_1-42_ and an increased risk of infections in immunocompetent β-secretase knockout mice were documented [68]. In the same manner, an increased rate of infections in AD patients treated with the Aβ_1-42_-lowering agent tarenflurbil has been reported [69]. More directly, Soscia et al. [40] discovered that Aβ_1-40_ and Aβ_1-42_ exert in vitro antimicrobial activity against eight common microorganisms with a potency equivalent to, and in some cases greater than, LL-37. In addition, the authors found an Aβ-mediated activity against yeast in brain homogenates of AD patients. The antimicrobial properties of Aβ were also confirmed by Spitzer et al. [70] which demonstrated that Aβ_x-42_ variants, but not Aβ_x-40_ variants, can bound to microbial surfaces and induce microbial agglutination. In addition, Aβ_x-42_ killed up to 80% of microorganisms in all tested pathogens (i.e., bacteria and yeast), whereas Aβ_1-40_ only had a moderate anti-yeast activity. To summarize, these results are consistent with the protective Aβ activity as AMP against pathogens that, when dysregulated, could lead to AD pathology.

More specifically than the Bioflocculant Hypothesis [66] and the Amyloid Protection Hypothesis [65], Moir et al. [22] proposed the Aβ “Anti-microbial Protection Hypothesis”. In line with the studies above discussed, the authors suggested that Aβ may play a function as an AMP; thus, representing a type of innate immune defense peptide that protects the host against a variety of pathogens [22]. The persistent activation of this pathway could lead to chronic inflammation and neurodegeneration in AD [22]. 

In addition to the high concentration of total tau (T-tau) and P-tau, the reduced levels of Aβ_1-42_ represent the third core CSF biomarkers for AD [71], whereas the ability to discriminate AD from non-AD patients based on the blood levels of Aβ_1-42_ remains unclear [72]. Indeed, a recent literature review showed that the salivary level of Aβ_1-42_ could represent a worthy candidate biomarker for the diagnosis of AD [73]. 

### 3.2. Lactoferrin 

Lactoferrin, first identified in 1939 in bovine milk [74], was subsequently isolated and purified from human and bovine milk [75,76]. Human lactoferrin is a glycoprotein of 691 amino acids, synthesized and secreted following induction by many exocrine glands of the body [77,78]. Beyond milk, it is expressed in several biological fluids such as saliva, tears, seminal fluid, and cerebrospinal fluid (CFS) [73,78]. In addition, it is expressed both by neurons and glial cells [79]. Lactoferrin exerts a wide range of physiological functions including iron binding/transferring, antioxidant activities, neuroprotective properties, regulation of the immune response, anti-inflammatory and anti-carcinogenic potential [78]. 

The antimicrobial proprieties of lactoferrin are conferred by its highly positive charged N-terminal region [80] which ensures that it can provide first line of defense against bacteria, viruses, fungi, free radicals, protozoa, and yeasts [80,81,82,83,84]. 

Interestingly, lactoferrin has been shown to bind Aβ [85,86] and detected in high concentration in neurons and glial cells [79], Aβ senile plaques, and neurofibrillary tangles [79] of the AD brains. In particular, Osmand and Switzer [87] found that lactoferrin is a constituent of Aβ senile plaques and neurofibrillary tangles of the limbic system in brain tissues of post-mortem AD. Kawamata et al. [79] extended these results by showing that lactoferrin is highly expressed and upregulated in both neurons and glial cells (astrocytes, oligodendrocytes, and microglia) of the brain tissues of AD patients compared to normal controls. In addition, the authors find that its expression increases with age and colocalizes with Aβ senile plaques and neurofibrillary tangles of nearly all AD-affected areas, most notably the hippocampus, angular cortex, and entorhinal cortex. Despite these promising results, only 16 years later another research group investigated and better characterized the role of lactoferrin in AD [88]. In detail, An et al. [88] analyzed the expression and localization of lactoferrin transcript in the cerebral cortex of AD and normal controls using real-time polymerase chain reaction (RT-PCR) and in situ hybridization. The results showed greater expression of lactoferrin mRNA in the cortical neutrophilic leukocytes of AD patients, compared to the control group. Given that neutrophilic leukocytes are localized in the activated microglia, the increased release of lactoferrin could occur during the inflammatory process in AD. 

In light of the infectious hypothesis of AD, these results suggest that, in this pathology, lactoferrin is synthesized and released mainly from activated microglia, in an attempt to counteract the accumulation of Aβ. Intranasal administration of human lactoferrin in the transgenic mouse model of AD (APPswe/PS1DE9) has been shown to promote the non-amyloidogenic metabolism of APP processing through activation of *α*-secretase and ADAM10, leading to the production of soluble form of APP, sAPPα, having a neuroprotective role. Indeed, sAPPα reduces generation and deposition of Aβ and improves spatial and cognitive learning ability in AD mice [89]. 

Recently, the potential role of lactoferrin in AD treatment has also been tested on human subjects. Fifty AD patients were randomly assigned into two age- and sex-matched groups that received either standard therapy (group 1, AD patients without lactoferrin) or lactoferrin capsules for three months. Results show that the administration of lactoferrin significantly improved cognitive functions, increased the serum levels of acetylcholine, serotonin, antioxidant, and anti-inflammatory markers and the expression of Akt in peripheral blood lymphocytes (PBL), as well as PI3K, and p-Akt levels in PBL lysate. In addition, the treatment with lactoferrin reduces the levels of key players of inflammation and oxidative stress involved in AD pathology (e.g., serum levels of Aβ_42_, cholesterol, oxidative stress markers, IL-6, HSP-90, caspase-3, P-tau, tau, MAPK1, and PTEN) probably modulating the p-Akt/PTEN pathway [90]. Despite these promising results, further studies are needed to confirm and better characterize the efficacy of lactoferrin in the treatment of AD as well as to explore its administration in the prevention of AD. 

Beyond treatment, the pioneering studies performed by Carro et al. [41,91] on the Spanish population, suggested that salivary lactoferrin could represent a useful diagnostic tool for AD. In the first study, the authors compared the salivary levels of lactoferrin between amnestic mild cognitive impairment (aMCI) patients (*n* = 15), AD patients (*n* = 36), and a cognitively healthy control group (*n* = 40). Results showed that the salivary lactoferrin levels were significantly reduced in aMCI and AD patients compared with the healthy control group. The decreased lactoferrin concentration was also correlated with MMSE score and the APOE ε4 allele status in patients with aMCI/AD and negatively associated with the stage of disease (aMCI and AD). Using linear regression and ROC analysis, the authors established a cutoff value of 7.43 mg/mL to discriminate aMCI/AD from healthy subjects with a sensitivity and specificity of 100%. This cutoff value was also tested and successfully used to classify another blinded cohort of aMCI, AD, and healthy control subjects. In addition, in a 56-subject AD subcohort the authors found that saliva lactoferrin significantly correlates with CSF Aβ_1-42_ and CSF T-tau compared to the control group (*n* = 68). To evaluate whether the reduced concentration of lactoferrin was specific to AD, the authors compared its levels between a cohort of PD subjects (*n* = 59) and a control group, finding significantly increased levels in the first group. Lastly, the authors also collected evidence on the possibility of predicting the development of aMCI/AD in healthy subjects based on salivary levels of lactoferrin. In particular, they recruited two different cohorts: 116 “nonclinical” and 190 apparently neurologically healthy subjects. Using the previously identified cutoff value, the authors classified 18 subjects with abnormally reduced lactoferrin levels (<7.43 mg/mL) and 288 with normal/high lactoferrin levels (>7.43 mg/mL). From 1 to 5 years later, 14 of 18 subjects had converted to a clinical diagnosis of aMCI or AD, whereas none of the subjects with a negative test value had converted to aMCI or AD. Thus, salivary lactoferrin levels appear to be also a useful tool for early identification of individuals at risk of developing aMCI/AD with a sensitivity of 100% and a specificity of 98.6% and thus more accurately than Aβ_1-42_ and T-tau in CSF [41]. To better understand whether the decreased salivary lactoferrin levels are specific to AD and thus suitable for its diagnosis, the same research group performed a second study in which the relationship was examined between salivary lactoferrin and cerebral Aβ load in patients with aMCI, AD, frontotemporal dementia (FTD)—as an example of another type of dementia—and a healthy control group [91]. Data showed that salivary levels were decreased only in aMCI/AD and were associated with amyloid–PET imaging profile; thus, supporting the possible use of this biomarker in the differential diagnosis of AD vs. FDT with a sensitivity and specificity over 87% and 91%, respectively [91]. However, Gleerup et al. [73] attempted to validate the use of salivary lactoferrin to discriminate AD from non-AD patients in the Danish population. In addition, this study was the first to evaluate the diagnostic potential of CSF levels of lactoferrin. Participants were divided into four different groups: healthy subjects (*n* = 20), MCI (*n* = 56), AD (*n* = 71), and non-AD patients (*n* = 75). The latter group included a heterogeneity of conditions such as vascular dementia (VaD), mixed dementia, FTD, dementia with Lewy bodies (DLB), and Parkinson’s disease with dementia (PDD). The results of this study showed that there were no statistically significant differences in the levels of CSF and salivary lactoferrin between the different groups. In addition, no significant relationships were found between lactoferrin and the CFS concentration of well-established dementia biomarkers (Aβ_1-42_, P-tau, and T-tau). However, given the small sample size and the extreme heterogeneity of the control group, it could be useful in future studies to increase the sample and make a comparison between salivary levels of lactoferrin in AD patients and—separately—with other neurodegenerative diseases (e.g., AD vs. FTD vs. VaD vs. DLB vs. PDD). In addition, given its role in iron transport, it would be interesting to investigate whether lactoferrin also plays a role in the well-documented iron accumulation in neurons of AD patients. 

### 3.3. Defensins

Defensins are cationic and small AMPs mainly expressed by microglia, astrocytes, and choroid plexus epithelial cells [92]. Based on their structures, they are commonly classified into three groups: α-defensins, β-defensins, and θ-defensins [93]. An increasing line of evidence indicated that α- and β-defensins can be considered as good biomarkers for AD diagnosis. In particular, the levels of α-defensins 1–2 appear to be high in saliva, blood, serum, and CFS of AD patients [43,94,95], the levels of α-defensin 3 in saliva, serum, and CFS [43,94,95], whereas the level of α-defensin 4 only in saliva [43]. In the same manner, an increased level of β-defensin 2 in the serum and CFS of AD patients has been reported compared to healthy controls [95]. 

Moreover, defensins seem to be also involved in the molecular mechanism of AD pathogenesis. Williams et al. [44] found an increased expression of β-defensin 1 within granulovacuolar degeneration structures localized in the cytoplasm of hippocampal pyramidal neurons and in astrocytes of AD compared to non-AD control brain. A higher level of both β-defensin 1 and β-defensin 1 mRNA was also observed in the choroid plexus of the AD brain. Interestingly, the increased iron deposition in AD may contribute to the elevated expression of β-defensin 1 within the choroid plexus. Overall, these findings suggest an active role for β-defensin 1 as a potential modulator of the host innate immune response within the central nervous system. Moreover, compared to control people, AD patients show a higher copy numbers polymorphism of the DEFB4 gene—that encodes for β-defensin 4 and influences the production of β-defensin 2—thus, explaining the increased levels of β-defensin 2 reported in serum and CFS of AD patients [95]. More recently Zhang et al. [45] proposed the “anti-amyloid and antimicrobial hypothesis” of AD which postulates that α-defensins can be considered as multi-target inhibitors to prevent both microbial infection and amyloid aggregation underlying the onset of AD. In support of this hypothesis, the authors found that some α-defensins contain β-rich structures that allow it to cross-interact with Aβ. This binding would seem to prevent the formation of amyloid plaques and to reduce amyloid-induced cell toxicity. Indeed, β-defensins retain their original antimicrobial activity upon the formation of complexes with Aβ. 

Although further investigations are needed, these findings open new scenarios for understanding the pathogenesis of AD and underline the therapeutic potential of AMP for amyloid diseases.

### 3.4. Cystatins 

Cystatins include a large superfamily of related proteins with several antimicrobial, antiviral, and immunomodulatory properties [96]. These proteins can be classified into three major categories: (i) Stefins (stefin A and B; also known as cystatin A and B); (ii) cystatins (cystatins C, D, S, SA, and SN); and (iii) kininogens [97]. Several lines of evidence suggested the involvement of cystatins in AD. First, cystatins A, B, and C colocalized with Aβ senile plaques in AD patients [98,99,100]. Second, all three of these cystatins are considered potential Aβ-binding proteins in vitro and are capable of breaking down amyloid aggregates in cells [101]. Third, cystatin B can inhibit the fibrillization of Aβ in vitro [101]. Fourth, cystatin C can bind and inhibit Aβ oligomerization also in vivo [102,103]. Other findings indicated that both cystatin A and cystatin B are also two regulation factors of inflammation that can inhibit cathepsins [104]. In particular, cystatin B may play a protective role in AD through the inhibition of cathepsin B, a β-secretase enzyme that cleaved APP to synthesize Aβ fragments [43,105]. Interestingly, cathepsin B is overexpressed following chronic exposure to some bacteria producing an AD-like phenotype [106]. These data stress the urgency to investigate the interplay between cystatins and the chronic exposure to microorganisms in AD patients. 

Other evidence supporting the involvement of cystatins in AD derives from genomic studies. It an association between cystatin C gene polymorphism and an increased risk of developing AD has been reported (for a review see: [107]). In addition, a point mutation in the cystatin C gene causes a particularly dominantly inherited type of amyloidosis: the hereditary cystatin C amyloid angiopathy (HCCAA; [108]). 

Beyond the possible role of cystatins in the pathogenesis of AD, other studies suggest that they could also be considered good diagnostic biomarkers. Indeed, the levels of cystatin C are reduced in the CFS of AD patients [107], whereas the levels of cystatins A and B are increased in the saliva of AD patients [43]. 

The potential role of cystatins in the treatment of AD remains largely unexplored. However, preliminary studies indicated that cystatin C appears to be neurotoxic both in vivo and in vitro [109,110], suggesting that cystatins must be used in future therapeutic studies with a special precaution. 

### 3.5. Thymosin β4

Thymosin β_4_ (Tβ_4_) is a small multifunctional peptide containing 43 amino acids, which protects tissues against damage and promotes their regeneration [111]. It has been reported that in the central nervous system Tβ4 is mainly released by activated microglia to inhibit neuroinflammation [111]; thus, exerting antimicrobial activity [43]. Therefore, it is plausible to hypothesize that in AD Tβ4 may be released by activated microglia—together with other AMPs and other substances—to counteract the inflammation due to the Aβ accumulation. To our knowledge, the possible role of Tβ4 in the pathophysiology of AD has never been investigated and conflicting results have been obtained from the few studies that examined its potential role as a biomarker of AD. Le Pera et al. [112] found unaltered levels Tβ4 in the CFS of AD patients. On the other hand, Contini et al. [43] found increased levels of Tβ4 in the salivary of AD patients compared to a healthy control group. Further studies are needed to better clarify these aspects. 

### 3.6. LL37

LL37 is a cationic and small AMP that belongs to a group of major mammalian AMP named cathelicidin. It is released by several types of cells such as salivary glands, neutrophils, leukocytes [113], as well as neurons and glial cells in response to pathogens [114,115]. Interestingly, LL37 can also activate astrocytes and microglia to induce the glial-mediated neuroinflammation and thus may exert a role in the pathogenesis of AD [114]. In particular, it has been proposed that neurons, when injured, released LL-37 which in turn activates microglia and astrocytes. Consequently, microglia and astrocytes also release LL-37 which can cause the translocation of NFκB proteins to the nucleus by binding receptors such as FPRL-1, P2 × 7, and P2Y11. In turn, this process can lead to the expression and release of pro-inflammatory cytokines such as TNFα, IL-1, and IL-6, giving rise to a positive feedback mechanism which causes further destruction of neurons [115]. Moreover, in vitro data show that LL37 can bind to Aβ_1-42_ to modulate its ability to form the long and straight fibrils characteristic of AD. Thus, the balanced or unbalanced spatiotemporal expression of Aβ_1-42_ and LL37 could impact AD onset and progression [116]. Recently, it is has been designed using bioinformatics tools as an analog of LL-37, namely kLL-39, that would appear to have an enhanced antimicrobial activity and a reduced toxicity for the host cells [117]. In vitro studies are needed to investigate the antimicrobial effects of kLL-39 in AD. 

### 3.7. Histatin 1 and Statherin

Histatin 1 and statherin represent two salivary peptides that also exert antimicrobial activity [118,119]. Contini et al. [43] compared the salivary proteome of AD patients with a healthy control group, finding increased levels of these two AMPs in addition to α-defensins, cystatins A and B in AD patients. Thus, also histatin 1 and statherin can be viewed as interesting objects of interest for future research on AD. 

## 4. Conclusions 

The main aim of this review was to comprehensively characterize the role of AMPs in AD. The studies reported point out the importance of AMPs in the pathogenesis and diagnosis of AD, opening new and poorly or unexplored avenues for the treatment of this incurable neurodegenerative disorder. 

A strong line of evidence indicated that Aβ—in addition to the other documented physiological functions—can also act as an AMP; thus, representing an innate immune response targeted to fight and neutralize pathogens. However, the persistent activation of this pathway could lead to Aβ accumulation that in turn conduces chronic neuroinflammation and neurodegeneration. To counteract neuroinflammation, activated microglia and other glial and cellular sources, increased the synthesis and release of several AMPs (e.g., lactoferrin, defensins, cystatins, and thymosin β4). Preliminary evidence indicates that some of these AMPs can bind with Aβ to: (i) prevent the formation of Aβ amyloid plaques (e.g., lactoferrin, α-defensin); (ii) inhibit the oligomerization and fibrillization of Aβ (e.g., cystatins); (iii) reduce amyloid-deposition (e.g., lactoferrin) and amyloid-induced cell toxicity (e.g., α-defensin); (iv) attempt to destroy Aβ amyloid plaques (e.g., cystatins). Other AMPs (e.g., lactoferrin) reduce the levels of key players of inflammation and oxidative stress involved in AD pathology and can also promote the non-amyloidogenic metabolism of APP processing through activation of the *α*-secretase pathway, leading to the production of neuroprotective soluble sAPPα. However, the unbalanced levels of some AMPs (e.g., LL37) may be neurotoxic and thus negatively impact AD onset and progression (Figure 1). 

These data drive future research toward a better characterization of the molecular mechanisms of action of AMPs in AD and make them good candidates for the development and experimentation of new treatments. Moreover, the role of AMPs in the pathogenesis of AD could also be genetic, considering that at least cystatin C polymorphisms represent a well-documented risk factor for the development of AD. Therefore, future research on the role of AMPs in AD should also proceed in this direction.

Besides pathogenesis, the literature suggests that AMPs may also represent good diagnostic candidates for the identification of a panel of biomarkers capable of identifying AD, especially from a salivary source, and thus potentially capable of cutting down the expensive and lengthy current diagnostic process of AD (Table 1). However, future studies are also needed to better assess the diagnostic potential of AMPs in AD. 

## Figures and Tables

**Figure 1 antibiotics-11-00726-f001:**
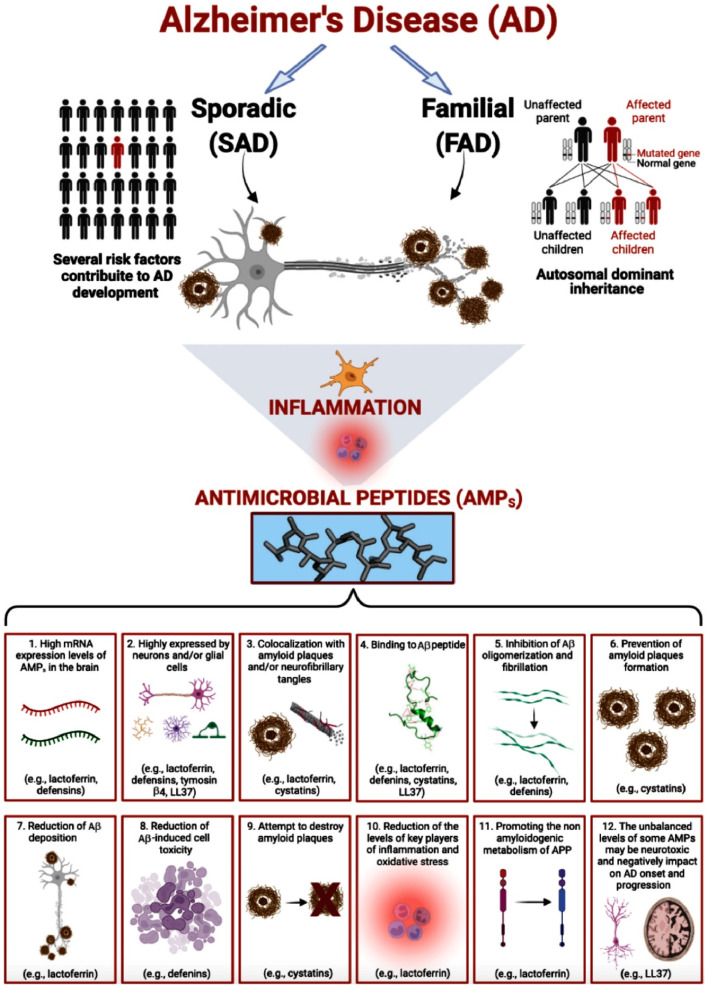
AMPs involvement in AD.

**Table 1 antibiotics-11-00726-t001:** AMP biomarkers in AD. Biomarkers are listed along with their source and relationship to AD.

Antimicrobial Peptide	Source	Description	Reference
Aβ_1-42_	Saliva	Increased in AD	[73]
	CFS	Reduced in AD	[71]
Lactoferrin	Saliva	Increased in AD	[41]
α-defensin 1	Saliva	Increased in AD	[43]
	Blood	Increased in AD	[94]
	Serum	Increased in AD	[95]
	CFS	Increased in AD	[95]
α-defensin 2	Saliva	Increased in AD	[43]
	Blood	Increased in AD	[94]
	Serum	Increased in AD	[95]
	CFS	Increased in AD	[95]
α-defensin 3	Saliva	Increased in AD	[43]
	Serum	Increased in AD	[95]
	CFS	Increased in AD	[95]
α-defensin 4	Saliva	Increased in AD	[43]
β-defensin 2	Serum	Increased in AD	[95]
	CFS	Increased in AD	[95]
Cystatin A	Saliva	Increased in AD	[43]
Cystatin B	Saliva	Increased in AD	[43]
Cystatin C	CFS	Decreased in AD	[107]
Thymosin β4	Saliva	Increased in AD	[43]
Histatin 1	Saliva	Increased in AD	[43]
Statherin	Saliva	Increased in AD	[43]

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
