# Peer review of "Antimicrobial Peptides (AMPs) in the Pathogenesis of Alzheimer’s Disease: Implications for Diagnosis and Treatment"

_antibiotics, 2022, doi:10.3390/antibiotics11060726_

Round 1

Reviewer 1 Report

the review is upto date. looks good.

Happy to see that the AMPs can be utilized for disease models as well.

Author Response

Thanks for your comment. We are happy that you enjoyed our work.

Best regards

Reviewer 2 Report

The authors present an interesting and comprehensive review of microbially-related etiologies of AD. Overall, the review reads well and is very clear. I only had a few formatting issues that need addressing prior to publication.

Line 126- The sentence beginning, "Several evidence suggested", should be reworded for subject-verb agreement.

Figure 1- Some of the images are not clear and lose meaning once the paper is printed. This could either be fixed by increasing the resolution of the figure or by providing a figure legend. There are also some English errors in the figure, for example, Highly mRNA expression levels... should likely be High mRNA expression levels.

Table 1- The references in the far right column need to be aligned.

Author Response

Comment. Line 126- The sentence beginning, "Several evidence suggested", should be reworded for subject-verb agreement.

Response. Thank you for this comment. We have now corrected this sentence.

Comment. Figure 1- Some of the images are not clear and lose meaning once the paper is printed. This could either be fixed by increasing the resolution of the figure or by providing a figure legend. There are also some English errors in the figure, for example, Highly mRNA expression levels... should likely be High mRNA expression levels.

Response. Thank you for this comment. We have inserted a higher quality figure and corrected any errors.

Comment. Table 1- The references in the far right column need to be aligned.

Response. Thank you for this comment. We have now aligned the reference in the far right column.

Reviewer 3 Report

In this review, Bruno et al. summarize the potential roles of antimicrobial peptides in Alzheimer’s disease (AD), from pathogenesis to diagnosis and treatment. Overall, the topic is interesting and very written. Even though, I have some comments to improve it.

The main comment is to remove the part of AMPs as biomarkers for AD diagnosis before the conclusion. And it is better to combine current sections 3-9, as sections 3.1-3.7, adding AMPs as biomarkers for AD as section 4.

Line 383, it is better not to show the first attempt, since literature reports such as PMID: 32849553 have compared the function of Aβ and AMPs, as well as their potential effect on AD. It is more correct to comprehensively characterize the role of AMPs in AD.

Other suggestions:

  1. Line 62, Antimicrobial Peptides (AMPs), no need to be italic and capitalized.
  2. The function of AMPs in autoimmune disease (e.g. PMID: 32457759) should be mentioned in lines 83-85.
  3. Line 158, Ab1-42 should be Aβ1-42.
  4. Line 185, a dash before In addition, the following sentence, Lactoferrin exert > exerts.
  5. Line 211, rephrase the sentence.

Author Response

Comment. The main comment is to remove the part of AMPs as biomarkers for AD diagnosis before the conclusion. And it is better to combine current sections 3-9, as sections 3.1-3.7, adding AMPs as biomarkers for AD as section 4.

Response. Thank you for this comment. We have now combined the section 3-9 as section 3.1.-3.7. However, it does not seem correct to us to make the part on biomarkers paragraph 4 since all this information is already reported in paragraphs 3.1-3.7. in the conclusions we simply summarized the information on AMPs as biomarkers, as well as those on AD pathogenesis and treatment, to follow the logic of the review reported in the title.

Comment. Line 383, it is better not to show the first attempt, since literature reports such as PMID: 32849553 have compared the function of Aβ and AMPs, as well as their potential effect on AD. It is more correct to comprehensively characterize the role of AMPs in AD.

Response. Thank you for this comment. We have replaced the sentence with “The main aim of this review was to comprehensively characterize the role of AMPs in AD”.

Comment. Line 62, Antimicrobial Peptides (AMPs), no need to be italic and capitalized.

Response. Thank you for this comment. We have now corrected this error.

Comment. The function of AMPs in autoimmune disease (e.g. PMID: 32457759) should be mentioned in lines 83-85.

Response. Thank you for this comment. We have now added this information and the reference you suggested.

Comment. Line 158, Ab1-42 should be Aβ1-42.

Response. Thank you for this comment. We have now corrected this error.

Comment. Line 185, a dash before In addition, the following sentence, Lactoferrin exert > exerts.

Response. Thank you for this comment. We have now corrected this error.

Comment. Line 211, rephrase the sentence.

Response. Thank you for this comment. We have now rephased the sentence.